# *Octorhopalona* *saltatrix*, a New Genus and Species (Hydrozoa, Anthoathecata) from Japanese Waters

**DOI:** 10.3390/ani12131600

**Published:** 2022-06-21

**Authors:** Sho Toshino, Gaku Yamamoto, Shinsuke Saito

**Affiliations:** 1Kuroshio Biological Research Foundation, 560 Nishidomari, Otsuki, Hata 788-0333, Japan; 2Enoshima Aquarium, Katasekaigan, Fujisawa 251-0035, Japan; g-yamamoto@enosui.com; 3Ibaraki Prefectural Oarai Aquarium, 8252-3 Isohamacho, Oarai, Higashi-ibaraki 311-1301, Japan; saitoh@aquaworld-oarai.com

**Keywords:** Halimedusidae, identification, phylogeny, new species, taxonomy

## Abstract

**Simple Summary:**

In this study, we describe a new genus and species of hydromedusa belonging to the family Halimedusidae (Hydrozoa, Anthoathecata) that is found off Oarai, Sagami Bay, and Tosa Bay, Japan. This family comprises four species in three genera: *Halimedusa*, *Tiaricodon,* and *Urashimea*. The new genus and species can be distinguished from all other Halimedusidae species by genetic sequences and morphological characteristics. A comparative table of the primary diagnostic characteristics of the genus *Halimedusa* is provided. In addition, the diagnosis of Halimedusidae was modified. Jellyfish blooms cause serious problems in fishing, industry and public health. This paper contributed to the understanding of ecology and diversity of jellyfish on our planet.

**Abstract:**

Approximately 300 species of cnidarian jellyfish have been reported in Japanese waters. However, many specimens remain unidentified. In this study, taxonomic investigations, including morphological observations and molecular 16S phylogenetic analyses, were conducted on unknown specimens collected off Oarai, Sagami Bay, and Tosa Bay, Japan. The specimens have the following morphological characteristics: distinct peaks in jelly above the base of the manubrium, a red band on the manubrium, and cylindrical marginal bulbs, each with an abaxial ocellus that is common to the family Halimedusidae. However, the specimens can be distinguished from other Halimedusidae species by their eight radial canals, eight tentacles with numerous stalked nematocyst knobs, and eight nematocyst tracks on the exumbrella. Moreover, molecular phylogenetic analyses revealed that the Kimura two-parameter distance between the specimens and other *Halimedusa* species was 0.066–0.099, which is considered to represent intergeneric variability. Based on this result, we described it as a new species and established a new genus for taxonomic stabilization. We also emended the diagnostic characters of the family Halimedusidae owing to the establishment of the new genus. Halimedusidae comprises five species in four genera. This paper provides taxonomic keys for the identification of species in the family Halimedusidae.

## 1. Introduction

The hydrozoan family Halimedusidae is a small group that includes four species from three genera: *Halimedusa* Bigelow, 1916; *Tiaricodon* Browne, 1902; and *Urashimea* Kishinouye, 1910 [1,2,3]. Halimedusidae species are distributed in the shallow waters of a range of tropical, subtropical, and mild-temperature localities in the Pacific and Atlantic Oceans [1,4,5,6]. The species have sexual, planktonic medusae, and asexual benthic polyps in their life cycle [1,6,7]. Free-swimming medusae are liberated by the budding of small solitary polyps.

Historically, the taxonomy of the family Halimedusidae has been unclear because of the limited differences in morphological characteristics among the genera. The first identified Halimedusidae species, *Tiaricodon coeruleus* Browne, 1902, was described by Browne (1902) and classified in the family Polyorchidae [4]. Kishinouye (1910) and Bigelow (1916) described *Urashimea globosa* Kishinouye, 1910 (in Cladonematidae) and *Halimedusa typus* Bigelow, 1916 (in Pandeidae) [8,9], but *Urashimea* was later moved into the family Pandeidae by Uchida and Nagao (1961) [7]. Bouillon (1995) removed *Urashimea* to the family Pandeidae [10]. Arai and Brinckmann-Voss (1980) erected the family Halimedusidae in the suborder Filifera [11], and *H. typus* was moved from Pandeidae. However, Mills (2000) moved Halimedusidae to the suborder Capitata within Anthoathecata based on new information about the morphology of both the medusa and polyp phases, including their life cycle and cnidomes [1]. Additionally, Mills moved the genera *Tiaricodon* and *Urashimea* from Polyorchidae to Halimedusidae.

Halimedusidae are characterized as follows: a medusa usually with a low gastric peduncle and distinct subumbrellar pockets in jelly above the manubrium base; manubrium cruciform with basal perradial lobes; mouth quadratic to cruciform with lips lined by nematocysts; four radial canals, either with four perradial marginal tentacles or four perradial marginal tentacles and four interradial groups of tentacles, all hollow; gonads either on the manubrium or on the manubrium and perradial lobes; no mesenteries; cylindrical marginal bulbs with abaxial ocelli (after Bouillion et al., 2006) [2]; small, solitary polyp with a perisarcal base and a protective perisarcal spine above the hydranth; 3–8 (generally 4) oral capitate tentacles, with a few scattered cnidocysts along their length; and single medusa buds, just below the tentacles [12].

To date, two described Halimedusidae species, *Urashimea globosa* and *Tiaricodon orientalis* Yamamoto and Toshino, 2021, have been reported in Japanese waters [3,13]. In this study, 16 specimens of an unidentified Anthoathecata species were collected off Oarai, Sagami Bay, and Tosa Bay, Japan. Our morphological and molecular phylogenetic analyses suggest that this Anthoathecata species should be regarded as a new genus and species within the Halimedusidae family.

## 2. Materials and Methods

### 2.1. Collection and Fixing

Sixteen unidentified medusae were collected near the water surface (within approximately 1 m) at Oarai Fishing Port, Higashi-Ibaraki, Ibaraki Prefecture; Katase Fishing Port, Enoshima, Fujisawa, Kanagawa Prefecture; and Shimonokae Fishing Port, Tosashimizu, Kochi Prefecture, Japan (Figure 1) between 15 December 2008, and 7 May 2021. The medusae were captured in either a dipper (diameter, 17 cm; volume, 2000 mL) or a dip net (mesh size, approximately 0.5 mm). Eight specimens were fixed in 3% formalin seawater and deposited in the Ibaraki Nature Museum, Ibaraki, Japan (INM); the National Museum of Nature and Science, Tsukuba, Japan (NSMT); and the Kuroshio Biological Research Foundation, Kochi, Japan (KBF). Seven specimens were preserved in 99.5% ethanol until molecular analysis. One specimen was used to determine the abundance of nematocyst types.

### 2.2. Molecular Phylogenetic Analysis

The 16S rDNA gene was used for the molecular phylogenetic analysis because it can effectively discriminate between species in Hydrozoa [14,15,16]. In this study, an approximately 600 bp fragment of mitochondrial 16S rDNA was used for phylogenetic analysis. Genomic DNA was extracted from the 99.5% ethanol-preserved tissue of cultured specimens using the DNeasy Blood and Tissue Kit (QIAGEN, Hilden, Germany) according to the manufacturer’s instructions. The 16S rDNA was PCR-amplified and sequenced with the forward and reverse primer pair TCGACTGTTTACCAAAAACATAGC and ACGGAATGAACTCAAATCATGTAAG [17], respectively, using the following PCR profile: an initial denaturation at 94 °C for 5 min; five cycles at 94 °C for 50 s, 45 °C for 50 s, and 72 °C for 60 s; 30 cycles at 94 °C for 50 s, 50 °C for 50 s, and 72 °C for 60 s; and a final elongation at 72 °C for 5 min [14]. The PCR products were purified using a QIAquick PCR Purification Kit (Qiagen, Germany) and sequenced in both directions using an ABI 3730 automated sequencer (Applied Biosystems, Bedford, MA, USA). The new sequences were aligned using MEGA 6.06 with built-in ClustalW [18]. Phylogenetic analysis and pairwise distance measurements were performed using the maximum likelihood method based on the Kimura 2-parameter model [19], with 1000 bootstrap replications in MEGA 6.06. Six sequences were deposited in GenBank under accession numbers LC653016-653021 for the new genus (Table 1) [14,20,21,22,23].

### 2.3. Morphological Investigation

Taxonomic observations and measurements were performed on live and preserved specimens (Figure 2). Measurements were made using ImageJ [24] to the nearest 0.1 mm. For nematocyst identification in the medusae, squash prepared from fresh tissues was examined under a compound microscope (ECLIPSE Ci, Nikon, Tokyo, Japan). Nematocysts were identified according to a previously described method [1,25]. To determine the abundance of nematocyst types in medusae, approximately 100 nematocysts were identified, measured, and counted in unregistered specimens. Measurements were performed using ImageJ [24] to the nearest 0.1 µm.

## 3. Results

### 3.1. Molecular Phylogenetic Analysis

We sequenced six individuals of *Octorhopalona saltatrix* for the 16S rDNA fragments, in addition to 12 Anthoathecata taxa, for statistical analyses. The maximum likelihood tree constructed for the family Halimedusidae based on the 16S rDNA sequences (Figure 3) comprised four major clades formed in the suborder Capitata: (1) *Moerisia inkermanica*, (2) *Tiaricodon orientalis*, (3) *Urashimea globosa*, and (4) *Octorhopalona*. The family paraphyly of Halimedusidae, including clades 1–3 and 4, was evident in the 16S rDNA phylogenetic tree with high bootstrap values (92%), which supports the validity of the genus. Moreover, Halimedusidae includes *M. inkermanica*, which is presently classified in the family Moerisiidae.

The Kimura two-parameter distance was 0.094–0.099 between *Octorhopalona saltatrix* n. sp. and *Tiaricodon*, 0.066 between *Octorhopalona* and *Urashimea*, and 0.064–0.067 between *Tiaricodon* and *Urashimea* (Table 2). The distance was greater (0.093–0.189) between different *Octorhopalona* and other Anthoathecata families, Cladonematidae, Corynidae, Moerisiidae, Pandeidae, Polyorchidae, and Zancleidae.

### 3.2. Morphological Investigation

#### 3.2.1. Systematics

Phylum Cnidaria Hatschek, 1888.

Subphylum Medusozoa Petersen, 1979.

Class Hydrozoa Owen, 1843.

Subclass Hydroidolina Collins, 2000.

Order Anthoathecata Cornelius, 1992.

Suborder Capitata Kühn, 1913.

Family Halimedusidae Arai and Brinckmann-Voss, 1980.

Genus *Octorhopalona* gen. nov.

New Japanese name: Otohime-kurage-zoku.

#### 3.2.2. Genus Diagnosis

Halimedusidae, with eight nematocyst tracks in the exumbrella; eight radial canals; eight characteristic perradial and interradial blead-shaped ‘smooth peaks’ in the mesoglea between the radial canals; eight tentacles, with numerous stalked nematocyst knobs over their entire length; and tentacular bulbs swollen with an abaxial ocellus.

Type species. *Octorhopalona saltatrix* sp. nov. is designated here.

The genus name ‘*Octorhopalona*’ is taken from the Greek words ‘octo’ and ‘rhopalon’, meaning ‘eight’ and ‘club’, respectively; the gender is feminine. The name indicates that the medusa bears eight tentacles that look like eight clubs.

#### 3.2.3. Species Description

##### *Octorhopalona saltatrix* sp. nov.

New Japanese name. Otohime-kurage.

Materials examined. Holotype: INM-1-96244; Oarai Fishing Port, Higashi-Ibaraki, Ibaraki Prefecture, eastern Japan; 36°18′39.1″ N, 140°34′39.5″ E; 15 December 2008; collector: Shinsuke Saito. Paratypes: KBF-M30; Enoshima, Fujisawa, Kanagawa Prefecture, eastern Japan; 35°17′52.4″ N, 139°28′32.2″ E; 7 May 2021; collector: Gaku Yamamoto; KBF-M31, same locality as KBF-M30; 24 September 2018; collector: Gaku Yamamoto: KBF-M32; Shimonokae Fishing Port, Tosashimizu, Kochi, Japan; 32°51′42.96″ N, 132°57′34.80″ E; 30 November 2020; collector: Sho Toshino; NSMT-Co1802. Katase Fishing Port, Fujisawa, Kanagawa Prefecture, eastern Japan; 35°18′23.6″ N 139°28′51.6″ E; 14 July 2019; collector: Haruka Onishi; NSMT-Co1803; Enoshima, Fujisawa, Kanagawa Prefecture, eastern Japan; 35°17′52.4″ N, 139°28′32.2″ E; 9 November 2018; collector: Haruka Onishi; NSMT-Co1804, same locality as NSMT-Co1803; 22 November 2018; collector: Gaku Yamamoto; and NSMT-Co1805, same locality as NSMT-Co1803; 14 July 2019; collector: Gaku Yamamoto (Table 3).

Description. Mature medusae have bell-shaped umbrellas (Figure 4 and Figure 5A–C), 9 mm in height and 9 mm in diameter (Table 3). The umbrella apex is rounded and the mesoglea is thickened (Figure 4 and Figure 5A). Exumbrella are smooth and nematocysts are sparsely scattered. Nematocysts track eight, perradial and interradial, on the exumbrella about two-thirds of the umbrella height (UH). Manubrium hang in the umbrella cavity, quadrilateral bottom, and are light brown or translucent (Figure 6A). The extended manubrium length was approximately 2 mm and did not extend beyond the umbrella margin. The gonads cover the entire surface of the manubrium, except for the mouth lip. The mouth cruciform has four lips. The stomach has short, sac-like perradial and interradial lobes (Figure 6B). There are eight characteristic interradial blead-shaped ‘smooth peaks’ in the mesoglea between the radial canals, rising above the level of the radial canals (Figure 6A). There are eight radial canals and one circular canal (Figure 5B,C and Figure 6B,C). The velum is narrow, with a velarial width of 10% of the umbrella diameter (UD) (Figure 6C). The tentacular bulbs were swollen, each with a dark brown abaxial ocellus (Figure 6D). There were eight tentacles, with numerous stalked nematocyst knobs in their entire length, approximately the same length as live umbrella height (UH) (Figure 6E).

The smallest young medusa had a UH of 1.3 mm and a UD of 0.9 mm. The mesoglea at the apex of the exumbrella was thinner than that of adults (Figure 7A). Nematocyst tracks on the exumbrella were approximately two-thirds of the UH. The manubrium was short, thin, translucent to whitish, and approximately one-third the length of the UH. Its mouth was simple and circular. It possessed eight radial canals and a single circular canal. Eight interradial peaks were absent in the mesoglea between the radial canals (Figure 7B). The velum was wide, and the velarial width was 20% of the UD (Figure 7C). The tentacular bulbs were swollen, each with a red-brown abaxial ocellus. There were eight tentacles, pearl-string-like, and 1–3 white or white-brown nematocyst batteries aligned, one-half of the UH.

Cnidome. Two nematocyst types were identified and measured (Table 4, Figure 8). Exumbrella: stenoteles. Manubrium: desmonemes and stenoteles. Tentacle: desmonemes and stenoteles. Tentacle bulb: desmonemes and stenoteles.

Habitat and ecology. Medusae of *Octorhopalona saltatrix* appeared on the surface of shallow waters (5–10 m in depth) off Oarai (eastern Japan) during November and December; in Sagami Bay (western Japan) from May to November; and in Tosa Bay (Western Japan) during November. Polyps of the species have never been found in the wild; however, medusa budding seems to occur between spring and fall in Japanese waters. In some specimens, the mesoglea at the apical part was infested with flukes.

Etymology. The specific name ‘*saltatrix*’ is taken from the Latin word ‘*saltatrix*’, meaning ‘female dancer’. The gender is feminine. The name reflects the medusa swimming like a female dancer using its umbrella and eight tentacles.

## 4. Discussion

### 4.1. Molecular Phylogenetic Analysis

The paraphyly of Halimedusidae, including *Octorhopalona* and three species (*U. globosa*, *T. coeruleus*, and *M. intermanica*), was evident in the 16S rDNA phylogenetic tree with high bootstrap values (92%), which supports the validity of the new species. The maximum-likelihood tree includes *Moerisia inkermanica* (Moerisiidae), because *Tiaricodon* was placed in the family Moerisiidae [26]. The tree suggests that *M. inkermanica* may be derived from within Halimedusidae.

The Kimura two-parameter distance was 0.094–0.099 between *Octorhopalona saltatrix* n. sp. and *Tiaricodon*, 0.066 between *Octorhopalona* and *Urashimea*, and 0.064–0.067 between *Tiaricodon* and *Urashimea*. The distance was greater (0.093–0.189) between different *Octorhopalona* and other Anthoathecata families, Cladonematidae, Corynidae, Moerisiidae, Pandeidae, Polyorchidae, and Zancleidae. In the class Hydrozoa, the intergeneric variability was 0.06–0.24 [22]. Given the results of the K2P distances in this study, *Octorhopalona* was considered a genetically independent genus in the family Halimedusidae.

### 4.2. Morphological Investigation

A comparison of the key features of the Halimedusidae genera is presented in Table 5. *Octorhopalona saltatrix* can be distinguished from all other Halimedusidae species by the number of tentacles and radial canals, as well as the shape of the tentacles. All the species in the family Halimedusidae have a bell-shaped umbrella, interradial peaks in the subumbrella, a red band on the manubrium, and abaxial ocelli on the basal tentacle bulbs. There are four tentacles (*Tiaricodon* and *Urashimea*) [3,4,5,6,26,27,28,29,30,31], four of which are perradial, and ten to eleven are interradial (*Halimedusa*) [1,9]. The tentacles are moniliform (*Halimedusa* and *Tiaricodon*) and numerous stalked nematocyst knobs cover their entire length (*Octorhopalona* and *Urashimea*). All Halimedusidae species have four radial canals and four interradial peaks in the subumbrella; however, only the *Octorhopalona* has eight radial canals and eight interradial peaks in the subumbrella. Therefore, we propose to amend the family Halimedusidae as follows:

Family Halimedusidae Arai and Brinckmann-Voss, 1980, sens. emend.

Diagnosis. Anthoathecata, with four or eight radial canals, a low peduncle, and distinct interradial or interradial and perradial peaks in the jelly above the base of the manubrium; red band on the manubrium, gonads extending out from the manubrium as lobes below the upper portions of the radial canals, but without mesenteries; mouth with lips lined by a row of sessile nematocyst clusters; four perradial hollow tentacles, or four perradial tentacles and four interradial single tentacles, or groups of hollow tentacles; cylindrical marginal bulbs, each with an abaxial ocellus.

Cnidomes are important for identification purposes. Two types of nematocysts, stenoteles and desmonemes, have been examined in *Octorhopalona*, *Halimedusa*, and *Tiaricodon* [1]. *Halimedusa* bears microbasic euryteles (medusae) and isorhizas or anisorhizas (polyps) [1]. *Tiaricodon* sp. from New Zealand and Brazil bear heteronemes and microbasic euryteles, respectively [5,28]. Cnidomes of *Urashimea* have not yet been examined.

## 5. Conclusions

Our morphological and molecular phylogenetic analyses suggest that these specimens collected off Oarai, Sagami Bay, and Tosa Bay, Japan are a new genera and new species belonging to the family Halimedusidae. This species appears in shallow waters in Japan during May and December. Polyps of this species have never been found in the wild. Additional sampling is required to understand their ecology and development.

Three genera, *Halimedusa*, *Tiaricodon*, and *Urashimea* are currently classified in the family Halimedusidae, defined by [1] as: Anthomedusae with four radial canals, with low peduncle and distinct interradial peaks in jelly above base of the manubrium; gonads extending out from the manubrium as lobes below the upper portions of the four radial canals, but without mesenteries; quadratic mouth with lips lined by a row of sessile nematocyst clusters, with four perradial hollow tentacles or with four perradial tentacles and four interradial groups of hollow tentacles; and possessing cylindrical marginal bulbs each with an abaxial ocellus. However, *Octorhopalona saltatrix* has eight tentacles, eight radial canals, and four interradial and four perradial peaks. Therefore, we propose to amend the family Halimedusidae as discussed.

## Figures and Tables

**Figure 1 animals-12-01600-f001:**
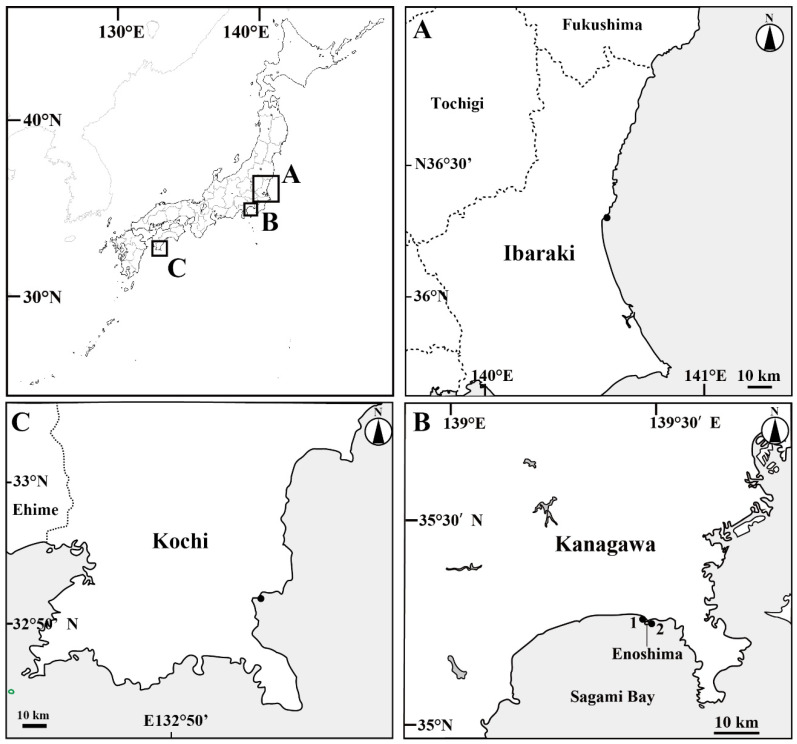
Map of the sampling sites. (**A**) Oarai Fishing Port, Higashi-Ibaraki, Ibaraki Prefecture; (**B**) Enoshima, Fujisawa, Kanagawa Prefecture: (1) Katase Fishing Port, (2) Enoshima; (**C**) Shimonokae Fishing Port, Tosashimizu, Kochi Prefecture. Black closed circles are sampling sites.

**Figure 2 animals-12-01600-f002:**
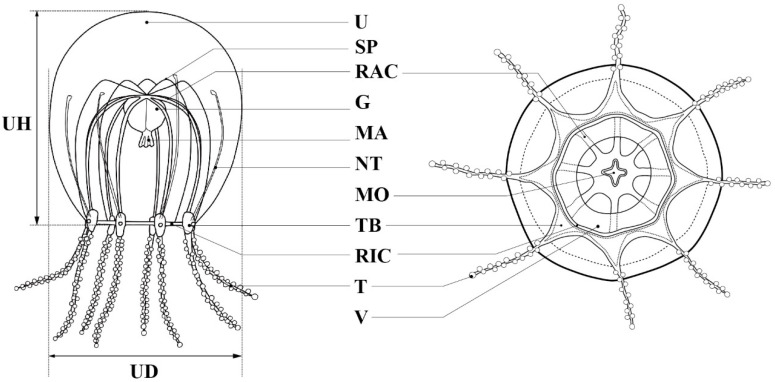
Key characters for identification and measurement of parts of the Halimedusidae: G = gonad; MA = manubrium; MO = mouth; NT = nematocyst track; RAC = radial canal; RIC = ring canal; SP = subumbrellar peak; T = tentacle; TB = tentacle bulb; U = umbrella; UD = umbrella diameter; UH = umbrella height; V = velum.

**Figure 3 animals-12-01600-f003:**
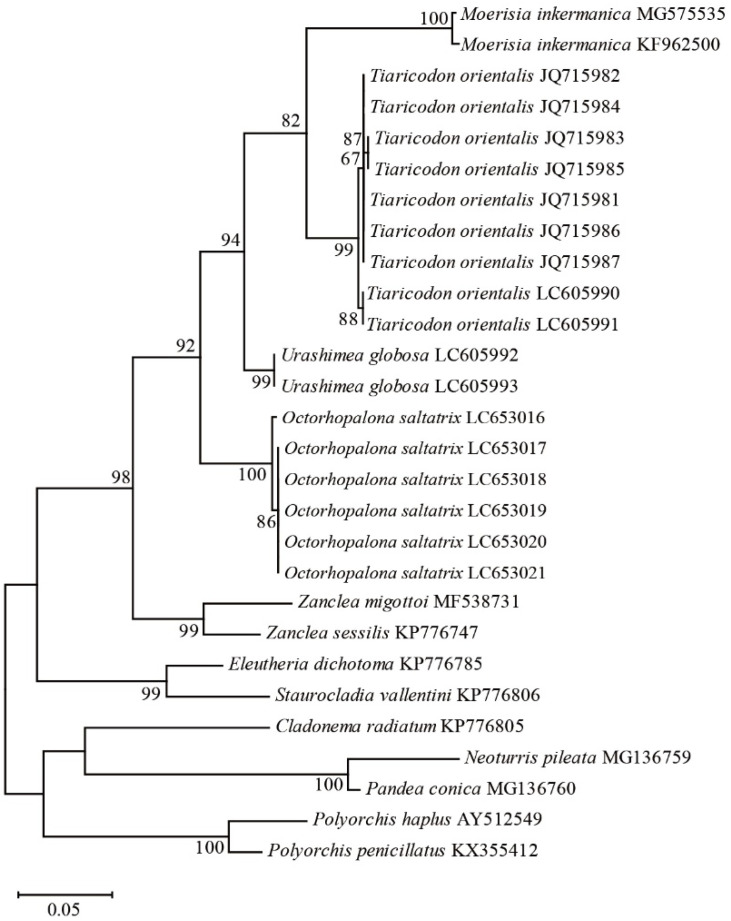
Nuclear 16S rDNA maximum-likelihood tree for 13 anthoathcata taxa based on the General Time Reversible model: Scale bar indicates branch length in substitutions per site. Nodal support values are presented as the ML bootstrap value; only values >50% are shown.

**Figure 4 animals-12-01600-f004:**
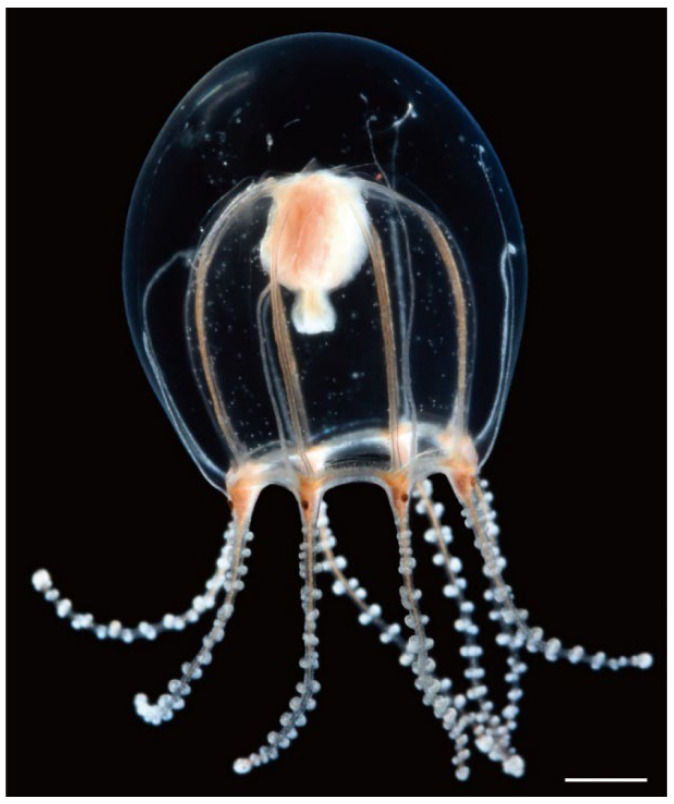
Live mature medusae of *Octorhopalona saltatrix* sp. nov. (unregisted specimen). Scale bar: 1 mm.

**Figure 5 animals-12-01600-f005:**
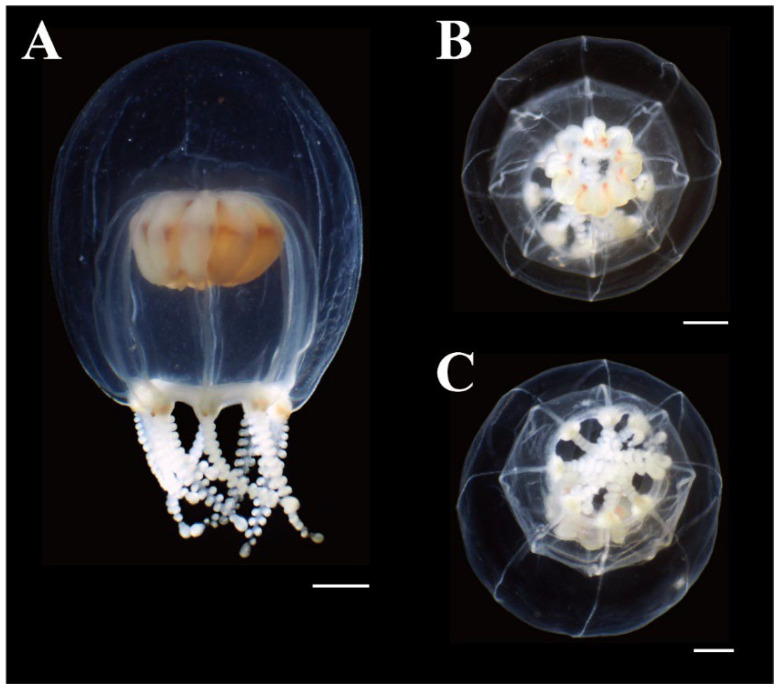
Preserved mature medusae of *Octorhopalona saltatrix* sp. nov. (NSMT-Co1802): (**A**) lateral, (**B**) apical, and (**C**) oral views. Scale bars: 1 mm.

**Figure 6 animals-12-01600-f006:**
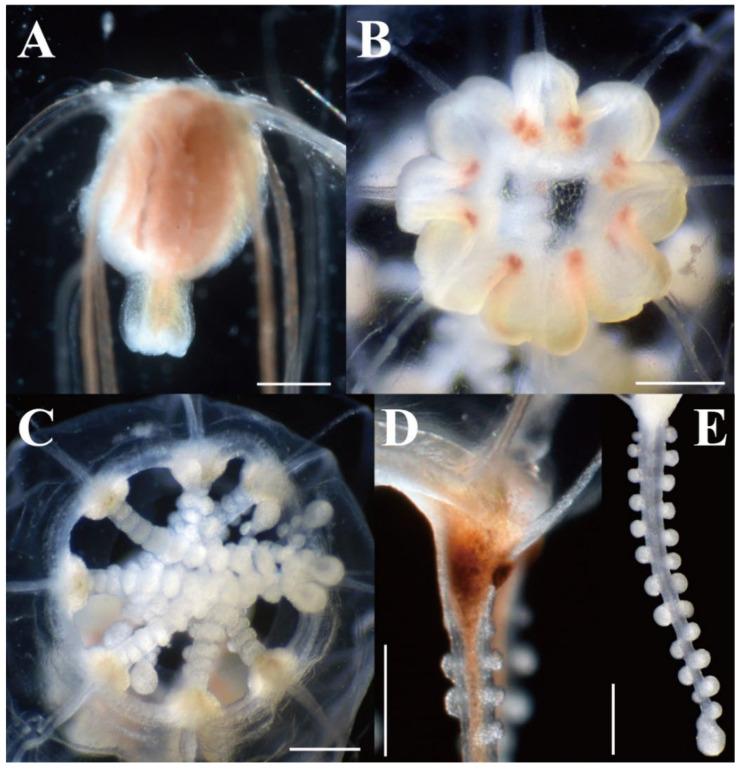
Live and preserved mature medusae of *Octorhopalona saltatrix* sp. nov. (unregisted specimen and NSMT-Co1802): (**A**) upper part of umbrella, (**B**) umbrella, apical view, (**C**) velum, (**D**) tentacular bulb, (**E**) tentacle. Scale bars: 0.5 mm.

**Figure 7 animals-12-01600-f007:**
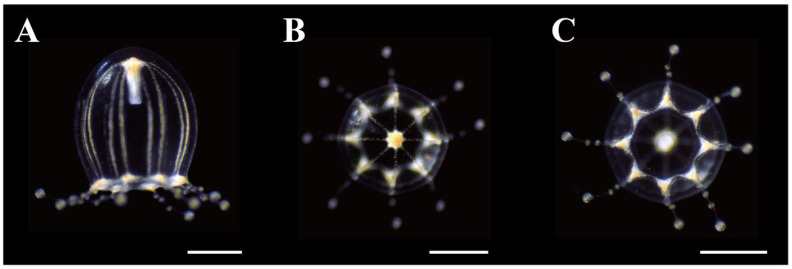
Live young medusae of *Octorhopalona saltatrix* sp. nov. (KBF-M32). (**A**) lateral, (**B**) apical and, (**C**) oral views. Scale bars: 0.5 mm.

**Figure 8 animals-12-01600-f008:**
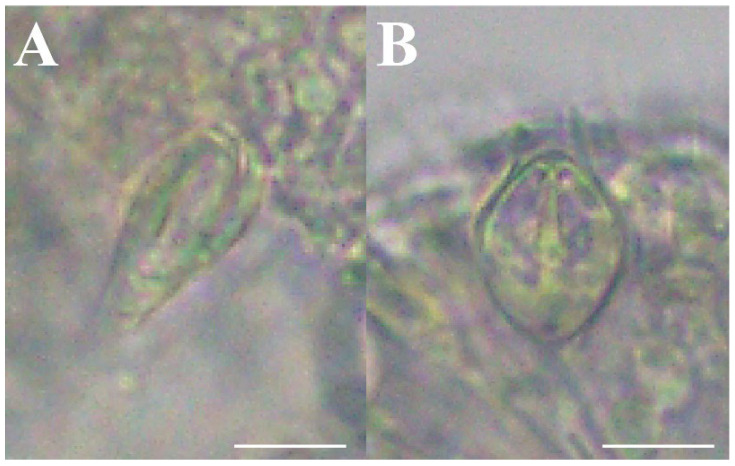
Nematocysts of *Octorhopalona saltatrix* sp. nov. (unregistered specimen): (**A**) desmoneme, (**B**) stenotele. Scale bars: 10 μm.

**Table 1 animals-12-01600-t001:** Taxa included in the phylogenetic analyses and GenBank accession numbers for the 16S rDNA sequences. Sequences obtained in this study are in bold.

Species	Accession No.	Locality (Origin)	Reference
*Cladonema radiatum*	KP776805	France: Roscoff, vivier	Unpublished
*Eleutheria dichotoma*	KP776785	Norway: Raunefjord	Unpublished
*Moerisia inkermanica*	KF962500	China	Unpublished
*Moerisia inkermanica*	MG575535	USA: Bayville, NJ	[20]
*Neoturris pileata*	MG136759	France: Bay of Villefranche-sur-Mer	[21]
** *Octorhopalona saltatrix* **	**LC653016**	Japan: Enoshima, Fujisawa, Kanagawa	This study
** *Octorhopalona saltatrix* **	**LC653017**	Japan: Enoshima, Fujisawa, Kanagawa	This study
** *Octorhopalona saltatrix * **	**LC653018**	Japan: Enoshima, Fujisawa, Kanagawa	This study
** *Octorhopalona saltatrix * **	**LC653019**	Japan: Enoshima, Fujisawa, Kanagawa	This study
** *Octorhopalona saltatrix * **	**LC653020**	Japan: Enoshima, Fujisawa, Kanagawa	This study
** *Octorhopalona saltatrix * **	**LC653021**	Japan: Enoshima, Fujisawa, Kanagawa	This study
*Pandea conica*	MG136760	France: Bay of Villefranche-sur-Mer	[21]
*Polyorchis haplus*	AY512549	USA: San Francisco Bay	[21]
*Polyorchis penicillatus*	KX355412	USA: Friday Harbour	[14]
*Staurocladia vallentini*	KP776806	Chile: Region de los Rios, Chaihuin	Unpublished
*Tiaricodon orientalis*	JQ715981	China	Unpublished
*Tiaricodon orientalis*	JQ715982	China	Unpublished
*Tiaricodon orientalis*	JQ715983	China	Unpublished
*Tiaricodon orientalis*	JQ715984	China	Unpublished
*Tiaricodon orientalis*	JQ715985	China	Unpublished
*Tiaricodon orientalis*	JQ715986	China	Unpublished
*Tiaricodon orientalis*	JQ715987	China	Unpublished
*Tiaricodon orientalis*	LC605990	Japan: Enoshima, Fujisawa, Kanagawa	[22]
*Tiaricodon orientalis*	LC605991	Japan: Enoshima, Fujisawa, Kanagawa	[22]
*Urashimea globosa*	LC605992	Japan: Tokoro Fishing Port, Kitami, Hokkaido	[22]
*Urashimea globosa*	LC605993	Japan: Tokoro Fishing Port, Kitami, Hokkaido	[22]
*Zanclea migottoi*	MF538731	Mexico: Gulf of Mexico, Alacranes Reef	[23]
*Zanclea sessilis*	KP776747	Spain: Mallorca, Cala Murada	Unpublished

**Table 2 animals-12-01600-t002:** Pairwise genetic distances (K2P) based on 440 positions of 16S sequences among Anthomedusae. The analysis involved 28 sequences.

No.		1	2	3	4	5	6	7	8	9	10	11	12	13	14	15	16	17	18	19	20	21	22	23	24	25	26	27
1	*Tiaricodon orientalis* JQ715981																											
2	*Tiaricodon orientalis* JQ715982	0.000																										
3	*Tiaricodon orientalis* JQ715983	0.002	0.002																									
4	*Tiaricodon orientalis* JQ715984	0.000	0.000	0.002																								
5	*Tiaricodon orientalis* JQ715985	0.002	0.002	0.000	0.002																							
6	*Tiaricodon orientalis* JQ715986	0.000	0.000	0.002	0.000	0.002																						
7	*Tiaricodon orientalis* JQ715987	0.000	0.000	0.002	0.000	0.002	0.000																					
8	*Tiaricodon orientalis* LC605990	0.005	0.005	0.007	0.005	0.007	0.005	0.005																				
9	*Tiaricodon orientalis* LC605991	0.005	0.005	0.007	0.005	0.007	0.005	0.005	0.000																			
10	*Urashimea globosa* LC605992	0.064	0.064	0.067	0.064	0.067	0.064	0.064	0.064	0.064																		
11	*Urashimea globosa* LC605993	0.064	0.064	0.067	0.064	0.067	0.064	0.064	0.064	0.064	0.000																	
12	*Octorhopalona saltatrix* LC653016	0.094	0.094	0.096	0.094	0.096	0.094	0.094	0.094	0.094	0.066	0.066																
13	*Octorhopalona saltatrix* LC653017	0.096	0.096	0.099	0.096	0.099	0.096	0.096	0.096	0.096	0.066	0.066	0.005															
14	*Octorhopalona saltatrix* LC653018	0.096	0.096	0.099	0.096	0.099	0.096	0.096	0.096	0.096	0.066	0.066	0.005	0.000														
15	*Octorhopalona saltatrix* LC653019	0.096	0.096	0.099	0.096	0.099	0.096	0.096	0.096	0.096	0.066	0.066	0.005	0.000	0.000													
16	*Octorhopalona saltatrix* LC653020	0.096	0.096	0.099	0.096	0.099	0.096	0.096	0.096	0.096	0.066	0.066	0.005	0.000	0.000	0.000												
17	*Octorhopalona saltatrix* LC653021	0.096	0.096	0.099	0.096	0.099	0.096	0.096	0.096	0.096	0.066	0.066	0.005	0.000	0.000	0.000	0.000											
18	*Moerisia inkermanica* KF962500	0.080	0.080	0.083	0.080	0.083	0.080	0.080	0.086	0.086	0.091	0.091	0.111	0.116	0.116	0.116	0.116	0.116										
19	*Moerisia inkermanica* MG575535	0.086	0.086	0.089	0.086	0.089	0.086	0.086	0.086	0.086	0.091	0.091	0.111	0.116	0.116	0.116	0.116	0.116	0.005									
20	*Zanclea migottoi* MF538731	0.134	0.134	0.137	0.134	0.137	0.134	0.134	0.134	0.134	0.111	0.111	0.111	0.116	0.116	0.116	0.116	0.116	0.149	0.149								
21	*Zanclea sessilis* KP776747	0.136	0.136	0.139	0.136	0.139	0.136	0.136	0.136	0.136	0.099	0.099	0.096	0.093	0.093	0.093	0.093	0.093	0.148	0.149	0.061							
22	*Eleutheria dichotoma* KP776785	0.161	0.161	0.164	0.161	0.164	0.161	0.161	0.161	0.161	0.151	0.151	0.151	0.157	0.157	0.157	0.157	0.157	0.170	0.170	0.155	0.154						
23	*Staurocladia vallentini* KP776806	0.186	0.186	0.190	0.186	0.190	0.186	0.186	0.183	0.183	0.158	0.158	0.154	0.160	0.160	0.160	0.160	0.160	0.170	0.164	0.157	0.154	0.069					
24	*Polyorchis haplus* AY512549	0.193	0.193	0.197	0.193	0.197	0.193	0.193	0.200	0.200	0.180	0.180	0.181	0.178	0.178	0.178	0.178	0.178	0.203	0.210	0.189	0.160	0.158	0.190				
25	*Polyorchis penicillatus* KX355412	0.168	0.168	0.171	0.168	0.171	0.168	0.168	0.168	0.168	0.155	0.155	0.153	0.159	0.159	0.159	0.159	0.159	0.174	0.174	0.174	0.164	0.149	0.183	0.051			
26	*Cladonema radiatum* KP776805	0.215	0.215	0.218	0.215	0.218	0.215	0.215	0.215	0.215	0.176	0.176	0.173	0.170	0.170	0.170	0.170	0.170	0.170	0.170	0.173	0.154	0.137	0.149	0.164	0.164		
27	*Neoturris pileata* MG136759	0.202	0.202	0.205	0.202	0.205	0.202	0.202	0.202	0.202	0.189	0.189	0.186	0.189	0.189	0.189	0.189	0.189	0.206	0.200	0.189	0.179	0.198	0.212	0.177	0.170	0.176	
28	*Pandea conica* MG136760	0.196	0.196	0.199	0.196	0.199	0.196	0.196	0.196	0.196	0.185	0.185	0.179	0.185	0.185	0.185	0.185	0.185	0.196	0.190	0.173	0.170	0.185	0.189	0.176	0.167	0.154	0.053

**Table 3 animals-12-01600-t003:** Morphometrics (mm) of *Octorhopalona saltatrix*. * The holotype. Nos. KBF-M30-32 and NSMT-Co1802-1805 are paratypes. UH, umbrella height; UD, umbrella diameter.

Specimen No.	UH (mm)	UD (mm)	Sampling Site	Date	Lat. Long.
INM-1-96244 *	6.4	5.8	Oarai Fishing Port, Oarai, Ibaraki, Japan	15 December 2008	36°18′39.1″ N, 140°34′39.5″ E
KBF-M30	4.8	4.7	Enoshima, Fujisawa, Kanagawa, Japan	7 May 2021	35°17′52.4″ N, 139°28′32.2″ E
KBF-M31	4.7	4.2	Enoshima, Fujisawa, Kanagawa, Japan	24 September 2018	35°17′52.4″ N, 139°28′32.2″ E
KBF-M32	2.4	2.9	Shimonokae Fishing Port, Tosashimizu, Kochi, Japan	30 November 2020	32°51′42.96″ N, 132°57′34.80″ E
NSMT-Co1802	6.4	5.5	Katase Fishing Port, Fujisawa, Kanagawa, Japan	14 July 2019	35°18′23.6″ N 139°28′51.6″ E
NSMT-Co1803	6.0	5.5	Enoshima, Fujisawa, Kanagawa, Japan	9 November 2018	35°17′52.4″ N, 139°28′32.2″ E
NSMT-Co1804	9.2	8.9	Enoshima, Fujisawa, Kanagawa, Japan	22 November 2019	35°17′52.4″ N, 139°28′32.2″ E
NSMT-Co1805	5.0	4.4	Enoshima, Fujisawa, Kanagawa, Japan	14 July 2019	35°17′52.4″ N, 139°28′32.2″ E

**Table 4 animals-12-01600-t004:** Cnidomes of *Octorhopalona saltatrix*. D, L represent capsule diameter and length, respectively, in μm.

Part	Type		Min	Max	Mean	SD	N
Exumbrella	Stenotele	D	4.8	7.4	6.0	0.6	31
		L	6.5	10.5	8.6	1.1	31
Manubrium	Desmoneme	D	2.4	3.4	2.9	0.3	8
		L	5.2	10.2	6.7	1.6	8
	Stenotele	D	5.4	7.3	6.0	0.7	7
		L	7.0	12.7	9.3	1.9	7
Tentacle	Desmoneme	D	3.6	5.6	4.7	0.6	12
		L	7.7	14.8	12.0	2.5	12
	Stenotele	D	4.2	10.2	5.7	1.2	20
		L	7.1	13.8	9.1	1.5	20
Tentacle bulb	Desmoneme	D	3.7	5.4	4.4	0.7	5
		L	9.8	14.1	11.7	1.6	5
	Stenotele	D	3.7	5.1	4.4	0.4	14
		L	5.3	6.5	5.8	0.3	14

**Table 5 animals-12-01600-t005:** Morphology of Halimedusidae medusa in previous and the present studies.

	*Octorhopalona* gen. nov.	*Halimedusa*	*Tiaricodon*	*Urashimea*
UH/UD (mm)	9/9	16/13	25/24	17/-
No. of radial canals	8	4	4	4
No. of peaks in subumbrella	8	4	4	many
No. of tentacles	4 perradial and 4 interradial tentacles	4 perradial tentacles and 4 clusters (up to 10–11) of interradial tentacles	4 perradial tentacles	4 perradial tentacles
Tentacle shape	Stalked nematocyst knobs	Moniliform	Moniliform	Stalked nematocyst knobs
Ocellus on tentacle bulbs	Present	Present	Present	Present
Umbrella	Bell-shaped	Bell-shaped	Bell-shaped	Bell-shaped or globular
Nematocyst tracks	Present	Absent	Absent	Present
Peduncle	Absent	Broad and low	Short, broad or indistinct	Absent
Red band on manubrium	Present	Present	Present	Present
Gonad	On manubrium with lobes extending out onto radial canals, as undulating swollen sacs	On manubrium with smooth lobes extending out onto radial canals (not folded or pendant sacs)	On manubrium with lobes extending out onto radial canals as swollen pouch-like sacs	On manubrium with lobes extending out onto radial canals, as undulating swollen sacs
Nematocysts	Stenoteles, desmonemes	Stenoteles, desmonemes, microbasic mastigophores, isorhizas or anisorhizas	Stenoteles, desmonemes, heteronemes	Unknown
Distribution	Off of Oarai; Sagami Bay; Tosa Bay, Japan	The Queen Charlotte Islands and the west coast of Vancouver Island in British Columbia; bays on the outer coast of Oregon; northern to central California	Sagami Bay, Japan; Falkland Islands, Wellington Harbor, New Zealand; Yellow Sea, Taiwan Strait, South China Sea, Xiamen Harbor; Cananeia-Iguape Coastal System, Sao Paulo, Brazil	Central to northern Japan, including Hokkaido, and on Sakhalin Island; Amoy, China; St. Helena Bay, South Africa
References	This study	[1,9]	[3,4,5,6,26,27,28]	[7,8,29,30,31]

## Data Availability

All datasets collected and analyzed during the current study are available from the corresponding author upon request.

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
