# Peer review of "Octorhopalona saltatrix, a New Genus and Species (Hydrozoa, Anthoathecata) from Japanese Waters"

_animals, 2022, doi:10.3390/ani12131600_

Round 1

Reviewer 1 Report

This is a very nice description of a new species in the family Halimedusidae. The documentation of the morphology and discussion of how it differs from other species in the family is excellent. The authors also generate molecular data (mitochondrial 16S; not nuclear as stated and not "near complete") for the new species and present a tree of this species and other relatively closely related taxa (including all available halimedusids). The authors make some small errors in interpreting the molecular results, but they will be easy to fix. The paper should be acceptable for publication after some minor revisions (mainly to the molecular section and some additional editing for clarity in the English).

There is no level of molecular divergence that would represent "inter-generic". That idea is a fallacy. It does not reflect an accurate understanding of evolution and phylogeny. Moreover, there are no exemplars of the genus Halimedusa. These could be more similar (and I would guess they will be, but who really knows?). The authors can describe a new genus if the morphology of the new species does not easily fit into any existing genus. 

The molecular analyses do NOT support monophyly of the family Halimedusidae. Instead, they suggest that the family is paraphyletic with respect to Moerisia. The authors can say that even without making a change to the classification of Moerisia. They can state that more work (more taxon sampling and more characters) should be conducted first.

Also, they should redo the molecular phylogenetic analysis with two changes: 1) include Odessia sequences (family Moerisidae) and 2) use software to pick the most appropriate model of nucleotide evolution.

I have included a marked up PDF with comments related to the above (as well as a few grammatical suggestions). I am very much looking forward to seeing this excellent work published. 

Author Response

Response to Reviewer 1 Comments

Comments and Suggestions for Authors

This is a very nice description of a new species in the family Halimedusidae. The documentation of the morphology and discussion of how it differs from other species in the family is excellent. The authors also generate molecular data (mitochondrial 16S; not nuclear as stated and not "near complete") for the new species and present a tree of this species and other relatively closely related taxa (including all available halimedusids). The authors make some small errors in interpreting the molecular results, but they will be easy to fix. The paper should be acceptable for publication after some minor revisions (mainly to the molecular section and some additional editing for clarity in the English).

There is no level of molecular divergence that would represent "inter-generic". That idea is a fallacy. It does not reflect an accurate understanding of evolution and phylogeny. Moreover, there are no exemplars of the genus Halimedusa. These could be more similar (and I would guess they will be, but who really knows?). The authors can describe a new genus if the morphology of the new species does not easily fit into any existing genus.

The molecular analyses do NOT support monophyly of the family Halimedusidae. Instead, they suggest that the family is paraphyletic with respect to Moerisia. The authors can say that even without making a change to the classification of Moerisia. They can state that more work (more taxon sampling and more characters) should be conducted first.

Also, they should redo the molecular phylogenetic analysis with two changes: 1) include Odessia sequences (family Moerisidae) and 2) use software to pick the most appropriate model of nucleotide evolution.

I have included a marked up PDF with comments related to the above (as well as a few grammatical suggestions). I am very much looking forward to seeing this excellent work published.

Response: We are grateful to an anonymous reviewer for the critical comments and useful suggestions that have helped us to improve our paper. As indicated in the responses that follow, we have taken all these comments and suggestions into account in the revised version of our paper.

Point 1: There is no magic number for intergeneric distances. I think this should be described as a new genus because it does not fit into any of the existing halimedusid genera.

Response 1: See response 16.

Point 2: has

Response 2: This error has been corrected in accordance with the reviewer's comment (p1, line 42). 

Point 3: was described by Browne (1902) and classified in the family Polyorchidae.

Response 3: We have changed in lines 46-47, page 2.

Point 4: eventually

Response 4: This error has been corrected in accordance with the reviewer's comment (p2, line 40).  

Point 5: Please note that Arai and Brinckmann-Voss moved just Halimedusa typus from Pandeidae when creating the family.

Response 5: We have changed in lines 50-51, page 2.

Point 6: The complete gene is much longer than 600bp, so this is not near-complete. Also, this gene is in the mitochondrial genome, not nuclear genome.

Can be stated as . . .

"An approximately 600 bp fragment of mitochondrial 16S rDNA was used for phylogenetic analysis."

Response 6: We have changed in lines 89-90, page 3.

 Point 7: Why was this model chosen? The authors could use something like ModelTest (JModelTest or Modeltest-NG) to select the most appropriate model of nucleotide evolution for this set of data.

Response 7: Thank you for your comment. But Kimura 2-parameter model also better model of nucleotide evolution. We will try to use these models in other our paper in the future. Therefore wish to retain the original text.

Point 8: The tree topology does not show Halimedusidae to be monophyletic. The clade includes Moerisia inkermanica. Instead, in this analysis Halimedusidae is shown to be paraphyletic with respect to Moerisia inkermanica, which is presently classified in the family Moerisiidae.

It would be quite interesting to add existing 16S data for Odessia maeotica to the analysis because this also is classified in Moerisiidae.

Response 8: Thank you for your constructive comment. We have changed in lines 131-135, page 5.

Point 9: between. "in" would refer to intraspecific variation which is much smaller.

Response 9: This error has been corrected in accordance with the reviewer's comment (p6, lines 138-139).  

Point 10: This sentence does not quite make sense. The tree showed that exemplars of each of the three Halimedusidae species cluster together in well-supported clades. In other words, each species is shown to be monophyletic, as expected.

Response 10: This sentence has deleted.

Point 11: This is not accurately stated. Halimedusidae is not shown to be monophyletic in this analysis. The clade with 92% bootstrap contains 4 species, 3 from Halimedusidae, plus Moerisia from Moerisiidae.

Response 11: We have changed in lines 257-258, page 12.

Point 12: No, this says nothing about the genus.

Response 12: We have changed in line 259, page 12.

Point 13: Tiaricodon is presently classified in Halimedusidae, even if it was once placed in Moerisidae.

Perhaps in total, you could state that your data indicate a close relationship between the two families and that indeed Moerisidae may be derived from within Halimedusidae. Actual taxonomic change (by including moerisiids in Halimedusidae) should wait until more complete analyses with both more taxa and more characters can be conducted.

Response 13: We have changed in lines 261-262, page 12.

Point 14: The tree indicates the exact opposite.

Response 14: We have changed in lines 261-262, page 12.

Point 15: in

Response 15: This errors have been corrected in accordance with the reviewer's comment (p12, lines 263-264).

Point 16: There is no good reason to expect that a particular divergence is indicative of different genera (or families, etc.). This new species should be placed in a new genus because its morphology does not fit with the existing halimedusidae genera.

Response 16: Thank you for your critical comment. The results of the K2P distance in this study was great between Octorhopalona and other Halimedusidae species. Therefore, we considered Octorhopalona as a new genus. We wish to retain the original text, but please let me know if you have any idea for revise the sentence.

Point 17: "All previously known Halimedusidae species have four. . . .".

Given that this new species has eight radial canals, you should provide a new diagnosis for the family. I think that should be here in the discussion, rather than the conclusion.

Response 17: We have moved a new diagnosis for the family from the conclusion (lines 280-290, pages 12-13.

Point 18: This should be in the discussion, rather than conclusion.

Response 18: See above comment.

Reviewer 2 Report

In the text the references must be indicated only by the number of the reference without writing authors and year (following the Author’s instructions of the Journal).

Author Response

Response to Reviewer 2 Comments

Comments and Suggestions for Authors

In the text the references must be indicated only by the number of the reference without writing authors and year (following the Author’s instructions of the Journal).

Response: Thank you for your comments. We have checked the number of the reference in the text. It seems no correction is needed.

Point 1: Halimedusa becomes Halimedusa Bigelow, 1916.

Response 1: We have add in line 38, page 1.

Point 2: Tiaricodon becomes Tiaricodon Browne, 1902

Response 2: We have add in line 38, page 1.

Point 3: Urashimea becomes Urashimea Kishinouye, 1910

Response 3: We have add in lines 38-39, page 1.

Point 4: add full species names (author and year)

Response 4: We have revised table 1.

Point 5: Verrill, 1865 becomes Hatschek, 1888

Response 5: We have revised in line 154, page 7.

Reviewer 3 Report

Dear Authors,

I found your manuscript very interesting and well-written. It's always fascinating to read about the description of new species and you have done it correctly and thoroughly with the right morphological and molecular comparisons. Just from the last aspects, I suggest giving more support to the use of the chosen gene, from the introduction to the discussion section. Some other clarifications and adjustments are needed in the material and methods and in the results sections. Please see below for the specific comments.

Materials and Methos

Paragraph 2.1: a more detailed list of the samples should be provided, considering the wide temporal window which includes them. Also, Figure 1 needs more care in better evidencing the sampling sites and linking the samples (from a list with coordinates or details) with the map in an appropriate way (also linking it to the successive Table 3).

Paragraph 2.2: why the 16S rDNA gene was used for the phylogenetic analysis? Could the authors provide some references to support this method in hydrozoan? Something about that must be included also in the introduction section to better support this choice.

Paragraph 2.3: the analysis in live or fresh specimens is in contrast with paragraph 2.1 in which the authors stated on about half of the samples were previously stored in formalin in a museum collection and the second half stored in ethanol for molecular analysis. Please address this discrepancy.

In 3.2.3 the species is described as Octorophalona saltatrix, but in the captions of Tables 3-4 and Figures  4-7, was reported as Octorophalona saltatorix. Please address this.

Discussion and Conclusion

The discussion section is in my opinion too synthetic but effective. Take into account my previous suggestion to better support the use of the 12S rDNA gene in your study, also in this section, to deepen this part with some comparisons with other related studies. On the contrary, the conclusion section is too long and comprises some discussion parts that I suggest moving in the previous section, leaving in the final one just a resume of your key results in a more synthetic way.

Have a nice work.

The Reviewer

Author Response

Response to Reviewer 3 Comments

Comments and Suggestions for Authors

I found your manuscript very interesting and well-written. It's always fascinating to read about the description of new species and you have done it correctly and thoroughly with the right morphological and molecular comparisons. Just from the last aspects, I suggest giving more support to the use of the chosen gene, from the introduction to the discussion section. Some other clarifications and adjustments are needed in the material and methods and in the results sections. Please see below for the specific comments.

Response: We are grateful to an anonymous reviewer for the critical comments and useful suggestions that have helped us to improve our paper. As indicated in the responses that follow, we have taken all these comments and suggestions into account in the revised version of our paper.

Point 1: Paragraph 2.1: a more detailed list of the samples should be provided, considering the wide temporal window which includes them. Also, Figure 1 needs more care in better evidencing the sampling sites and linking the samples (from a list with coordinates or details) with the map in an appropriate way (also linking it to the successive Table 3).

Response 1: The reviewer notes more detailed list of the samples should be provided. However, figure 1 and Table 3 provided enough information of sampling sites. Therefore wish to retain the original figure and table.

Point 2: Paragraph 2.2: why the 16S rDNA gene was used for the phylogenetic analysis? Could the authors provide some references to support this method in hydrozoan? Something about that must be included also in the introduction section to better support this choice.

Response 2: We have changed in lines 89-91, page 3.

Point 3: Paragraph 2.3: the analysis in live or fresh specimens is in contrast with paragraph 2.1 in which the authors stated on about half of the samples were previously stored in formalin in a museum collection and the second half stored in ethanol for molecular analysis. Please address this discrepancy.

Response 3: This error has been corrected in accordance with the reviewer's comment (p2, line 68, 74). 

Point 4: In 3.2.3 the species is described as Octorophalona saltatrix, but in the captions of Tables 3-4 and Figures  4-7, was reported as Octorophalona saltatorix. Please address this.

Response 4: This error has been corrected in accordance with the reviewer's comment (Tables 3-4, Figures 4-7).

Point 5: The discussion section is in my opinion too synthetic but effective. Take into account my previous suggestion to better support the use of the 12S rDNA gene in your study, also in this section, to deepen this part with some comparisons with other related studies. On the contrary, the conclusion section is too long and comprises some discussion parts that I suggest moving in the previous section, leaving in the final one just a resume of your key results in a more synthetic way.

Response 5: Thank you for your constructive comment. We have changed discussion section. We understand that 12S rDNA gene is used for molecular phylogenetic analysis due to effectively discriminates between species in Hydrozoa, but it is not fit in discussion section in our paper. We will try to use 12S rDNA in other our study.
